# Vaccine-Induced Antibody Responses against SARS-CoV-2 Variants-Of-Concern Six Months after the BNT162b2 COVID-19 mRNA Vaccination

Pinja Jalkanen,[a] Pekka Kolehmainen,[a] Anu Haveri,[b] Moona Huttunen,[a] Larissa Laine,[b] Pamela Österlund,[b] Paula A. Tähtinen,[c] Lauri Ivaska,[c] Sari Maljanen,[a] Arttu Reinholm,[a] Milja Belik,[a] Teemu Smura,[d] Hanni K. Häkkinen,[e] Eeva Ortamo,[e] Anu Kantele,[e] Ilkka Julkunen,[a,f] Johanna Lempainen,[c,f,g] Laura Kakkola[a]

[a]Institute of Biomedicine, University of Turku, Turku, Finland
[b]Finnish Institute for Health and Welfare, Helsinki, Finland
[c]Department of Paediatrics and Adolescent Medicine, Turku University Hospital and University of Turku, Turku, Finland
[d]Department of Virology, University of Helsinki, Helsinki, Finland
[e]Meilahti Infectious Diseases and Vaccine Research Center, MeiVac, Department of Infectious Diseases, University Hospital and University of Helsinki, Helsinki, Finland
[f]Clinical Microbiology, Turku University Hospital, Turku, Finland
[g]Immunogenetics Laboratory, Institute of Biomedicine, University of Turku, Turku, Finland

Anu Kantele, Ilkka Julkunen, Johanna Lempainen, and Laura Kakkola contributed equally to this article. Author order was determined by the corresponding author after negotiation.

**ABSTRACT** The emergence of severe acute respiratory syndrome coronavirus 2 (SARS-CoV-2) variants has raised concern about increased transmissibility, infectivity, and immune evasion from a vaccine and infection-induced immune responses. Although COVID-19 mRNA vaccines have proven to be highly effective against severe COVID-19 disease, the decrease in vaccine efficacy against emerged Beta and Delta variants emphasizes the need for constant monitoring of new virus lineages and studies on the persistence of vaccine-induced neutralizing antibodies. To analyze the dynamics of COVID-19 mRNA vaccine-induced antibody responses, we followed 52 health care workers in Finland for 6 months after receiving two doses of BNT162b2 vaccine with a 3-week interval. We demonstrate that, although anti-S1 antibody levels decrease 2.3-fold compared to peak antibody levels, anti-SARS-CoV-2 antibodies persist for months after BNT162b2 vaccination. Variants D614G, Alpha, and Eta are neutralized by sera of 100% of vaccinees, whereas neutralization of Delta is 3.8-fold reduced and neutralization of Beta is 5.8-fold reduced compared to D614G. Despite this reduction, 85% of sera collected 6 months postvaccination neutralizes Delta variant.

**IMPORTANCE** A decrease in vaccine efficacy against emerging SARS-CoV-2 variants has increased the importance of assessing the persistence of SARS-CoV-2 spike protein-specific antibodies and neutralizing antibodies. Our data show that after 6 months post two doses of BNT162b2 vaccine, antibody levels decrease yet remain detectable and capable of neutralizing emerging variants. By monitoring the vaccine-induced antibody responses, vaccination strategies and administration of booster doses can be optimized.

**KEYWORDS** COVID-19, coronavirus, vaccines

The spread of severe acute respiratory syndrome coronavirus 2 (SARS-CoV-2) began in December 2019 and expanded into a pandemic with the emergence of several genetic variants and evolutionary lineages of the virus. The World Health Organization (WHO) determines the variants with genetic changes that affect virus characteristics and have an apparent epidemiological impact as variants-of-interest (VOIs). The variants with

Address correspondence to Pinja Jalkanen, pinja.r.jalkanen@utu.fi.

The authors declare no conflict of interest.

increased infectivity, higher transmissibility, and ability to evade infection and vaccine-induced immunity are indicated as variants of concern (VOCs) (1, 2). In October 2021, WHO lists two VOIs (Lambda, C.37; and Mu, B.1.621) and four VOCs (Alpha, B.1.1.7; Beta, B.1.351; Gamma, P.1; and Delta, B.1.617.2). The U.S. government SARS-CoV-2 Interagency Group (SIG) has added one more classification, variants-being-monitored (VBM), referring to VOIs and VOCs that are no longer detected or are circulating in low amounts (3). SIG classifies only B.1.617.2 as VOC and all others, including also Epsilon (B.1.427 and B.1.429), Eta (B.1.525), Iota (B.1.526), Kappa (B.1.617.1, 1.617.3), and Zeta (P.2), as VBMs. Delta has become a globally dominant variant since its emergence in October 2020 (2, 4), causing an upsurge in cases with coronavirus disease (COVID-19) (5).

All VOCs have amino acid substitutions and deletions in the spike glycoprotein. Receptor binding domain (RBD) of the spike protein binds to the host cell receptor, angiotensin converting enzyme 2 (ACE2) (6), and RBD-specific antibodies correlate with the neutralization of the virus (7), thus raising the concern of amino acid changes in the RBD of the variants. Indeed, studies have shown decreased neutralization of Beta (B.1.351) by vaccine-induced antibodies (8–10). Also, preliminary results indicate a 5.8-fold decrease in neutralization titers against Delta (B.1.617.2) in BNT162b2 vaccinees 1 month postvaccination (10). Vaccine effectiveness studies on BNT162b2 mRNA COVID-19 vaccine have estimated 75–84% vaccine efficacy against Beta (B.1.351) infection (11, 12) compared to 90–94% efficacy against Alpha (B.1.1.7) (11, 13), 88% efficacy against Delta (B.1.617.2) (13), and 93% efficacy against non-VOC infection (12).

As a prerequisite for making decisions on vaccine booster doses and global prevention measures, it is crucial to analyze the persistence of antibodies in geographically and demographically diverse groups of vaccinees and assess the neutralizing capacity of antibodies against VOIs and VOCs. To elucidate this, we analyzed the spike protein-specific antibody levels among Finnish health care workers (HCWs) up to 6 months post BNT162b2 vaccination. We also analyzed the neutralization capacity of vaccinees' sera on two ancestral strains (Wuhan-like and D614G), three VOCs (Alpha, Beta, and Delta), and one VBM (Eta). Our data demonstrate a gradual decrease in anti-spike antibody levels and neutralizing antibody titers. Despite this decrease, 6 months after vaccination, 85-100% of vaccinees have neutralizing antibodies against Alpha and Delta variants.

## RESULTS

**Dynamics of SARS-CoV-2 spike-specific IgG antibodies 6 months after the BNT162b2 vaccination.** In Finland, the COVID-19 vaccinations started in December 2020 with two doses of BNT162b2 COVID-19 mRNA vaccine with a 3-week interval. HCWs received vaccines within the occupational health care. Serum samples were collected from HCWs selected from our previous study where we analyzed antibody levels up to 6 weeks postvaccination (8). Serum samples were obtained from 52 HCWs (67% females) aged 22 to 65 years (mean 45 years, median 47 years) before or up to 12 days after the first vaccine dose ("0d" sample) and 3 weeks (16–24 days, mean 20 days, standard deviation [SD] 1.5 days), 6 weeks (37–49 days, mean 43 days, SD 2.5 days), 3 months (83–106 days, mean 94 days, SD 6.7 days), and 6 months (154–262 days, mean 197 days, SD 31.6 days) after the first vaccine dose.

To analyze the longevity of IgG and total Ig antibodies after the BNT162b2 vaccination, the antibody levels were measured with enzyme immunoassay (EIA) against spike protein subunit S1 representing the original SARS-CoV-2 isolate Wuhan Hu-1 that is encoded also by the vaccine mRNA. As shown in our previous study (8), S1-specific IgG and total Ig antibody levels increased 3 weeks after the first vaccine dose and all vaccinated HCWs were seropositive at 6 weeks after the first vaccine dose (equal to 3 weeks after the second vaccine dose) (Fig. 1). Three and 6 months after the vaccination, anti-S1 IgG levels decreased with the 1.2- and 2.3-fold decrease, respectively, compared to the 6-week time point. Anti-S1 total Ig levels decreased 1.4- and 2.8-fold, respectively. Despite the decline, all vaccinees had S1-specific IgG antibodies 6 months after the vaccination (Fig. 1).

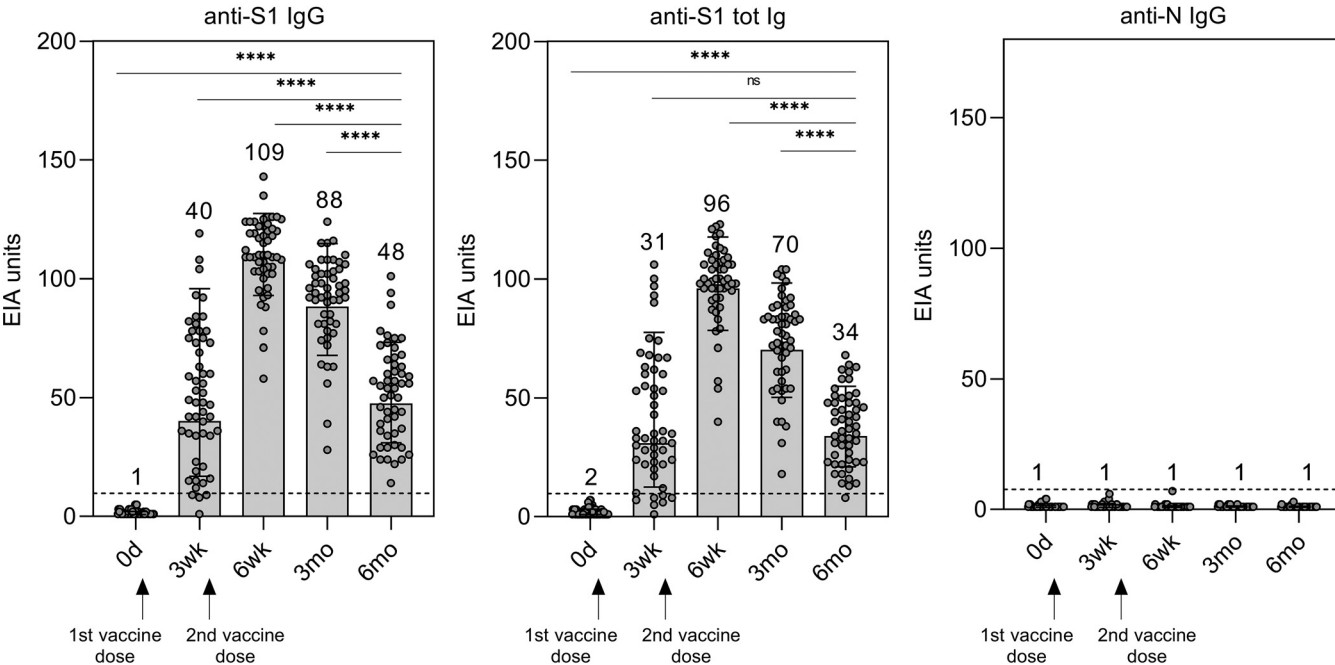

**FIG 1** Antibody responses in BNT162b2 vaccinated HCWs against SARS-CoV-2 spike glycoprotein subunit S1 and nucleoprotein (N) representing isolate Wuhan Hu-1. Anti-SARS-CoV-2 S1 and N IgG and anti-S1 total Ig antibody levels were analyzed with EIA. Serum samples were collected before vaccination, 3 weeks, 6 weeks, 3 months, and 6 months after the first vaccine dose. All vaccinees ($n$ = 52) received two doses of BNT162b2 vaccine at a 3-week dosing interval. Data are represented as geometric mean (GM) and geometric standard deviation (SD). GMs are indicated above each bar. Cut-off values are shown as dotted lines. Difference between time points was analyzed with Wilcoxon matched-pairs signed rank test using Pratt's method. Two-tailed $P$ values < 0.05 were considered statistically significant. Ns = not significant; ****, $P$ < 0.0001.

In addition, anti-SARS-CoV-2 nucleoprotein (N) IgG antibody levels were measured to identify prior SARS-CoV-2 infections and breakthrough infections. Based on anti-N IgG antibody levels, all vaccinees were seronegative at the time of the vaccination and did not contract SARS-CoV-2 infection during the 6-month follow-up (Fig. 1).

**SARS-CoV-2 variants and amino acid changes.** To determine the neutralization capacity of the vaccine-induced antibodies in microneutralization test with live viruses, SARS-CoV-2 variants representing five variants were isolated from Finnish COVID-19 patients. SARS-CoV-2 isolates FIN1-20, FIN25-20, FIN35-21, FIN33-21, FIN32-21, and FIN37-21 representing original Wuhan-like strain, ancestral D614G strain, and variants Alpha, Eta, Beta, and Delta, respectively, were propagated in cells and sequenced to determine the amino acid (aa) changes compared to Wuhan Hu-1 isolate (Fig. 2a). The aa substitutions and deletions of variants were mapped on the structure of SARS-CoV-2 trimeric spike protein (PDB: 6VXX) (Fig. 2b).

FIN1-20 (B) used in this study represented the original Wuhan-like strain despite H49Y substitution and ΔQTQTN675-679 deletion from cell culture adaptation in the spike protein. FIN25-20 (B.1) had D614G substitution along with ΔYQTQT674-678 and R682W changes near the furin cleavage site. Isolates representing Alpha (B.1.1.7), Eta (B.1.525), Beta (B.1.351), and Delta (B.1.617.2) variants had typical aa changes that define the lineages, and Alpha and Eta had R682W substitution from cell culture adaptation in all the sequence reads. Multiple substitutions were mapped to the sites that potentially affect the ACE2 binding (Fig. 2), including N501Y in both Alpha and Beta, E484K in Eta and Beta, K417N in Beta, and L452R and T478K only in Delta. In addition, all variants had deletions and/or substitutions in the N-terminal domain (NTD) of the spike protein known to contain epitopes for neutralizing antibodies (14).

**Neutralization of five SARS-CoV-2 variants up to 6 months after vaccination.** The ability of BNT162b2 vaccine-induced antibodies to neutralize SARS-CoV-2 variants D614G (B.1), Alpha (B.1.1.7; circulating in Finland during the first half of the year 2021), Eta (B.1.525), Beta (B.1.351), and Delta (B.1.617.2; circulating in Finland at the time of

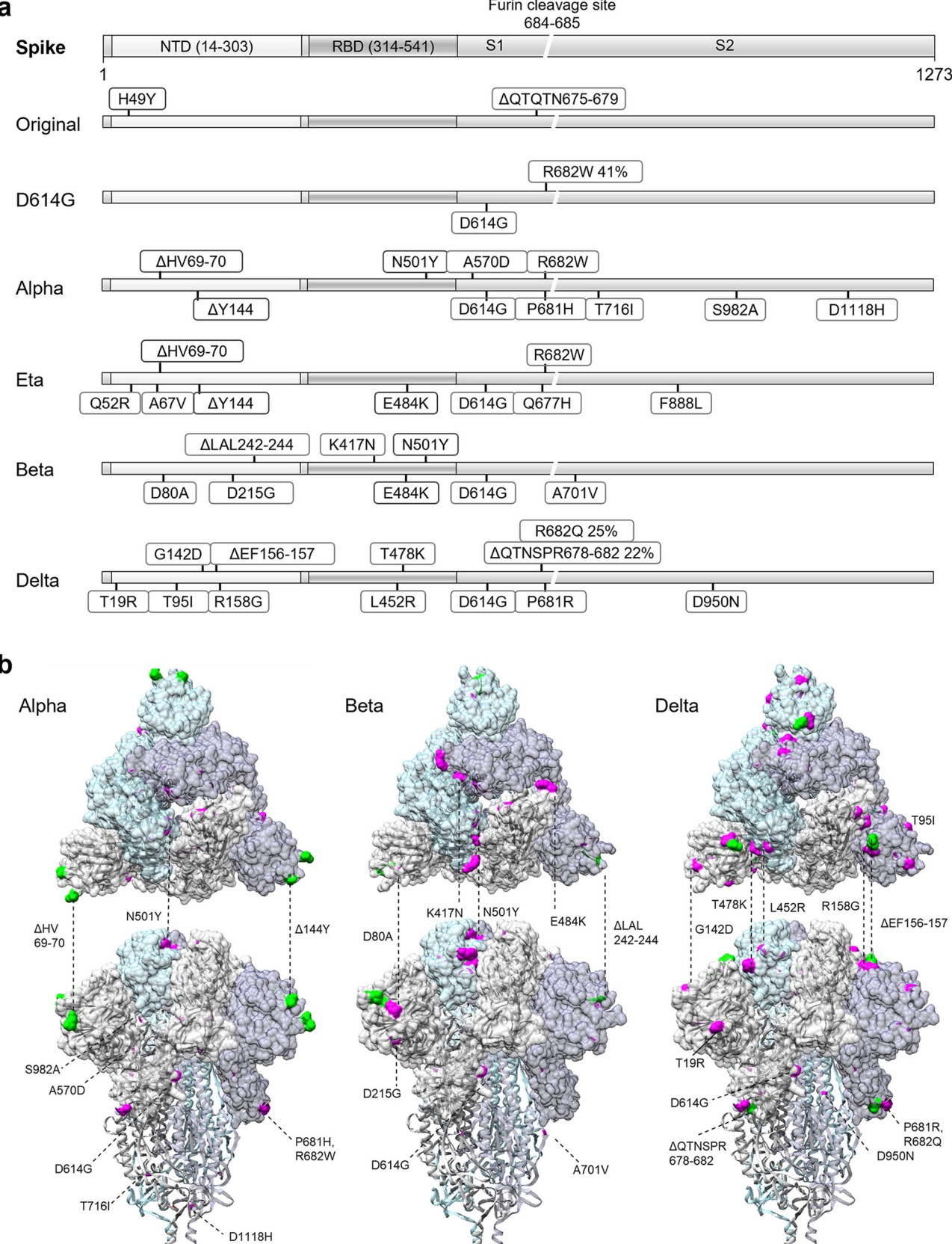

**FIG 2** Representation of genetic variants of SARS-CoV-2. (a) Schematic representation of SARS-CoV-2 spike protein. Amino acid substitutions and deletions present in over 20% of the NGS-obtained sequence reads are indicated for the variants used in this study: original Wuhan-like strain (B),

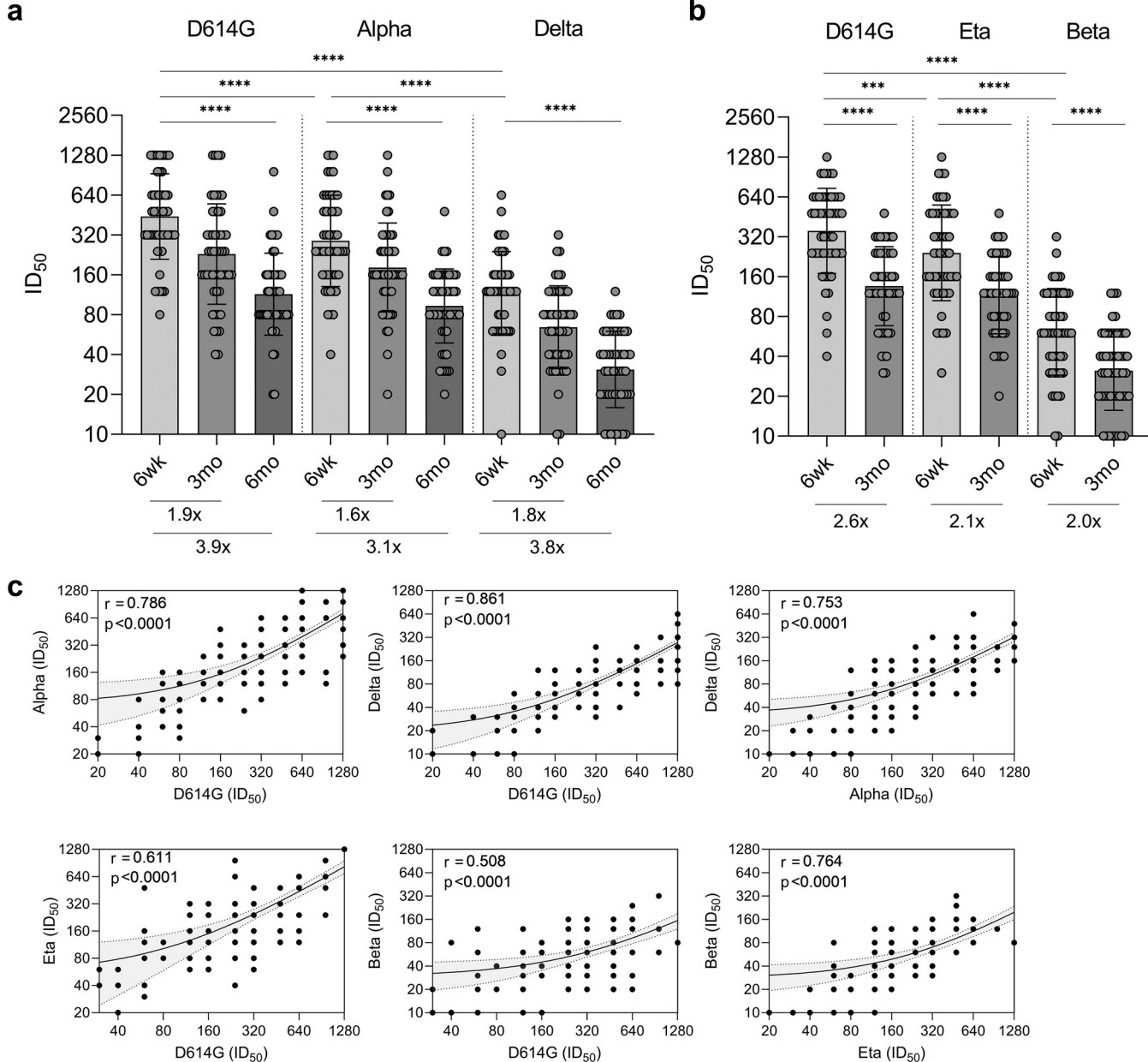

**FIG 3** Neutralization of five SARS-CoV-2 variants by sera of BNT162b2 vaccinated HCWs. Neutralization of variants (a) Alpha (B.1.1.7) and Delta (B.1.617.2) and (b) variants Eta (B.1.525) and Beta (B.1.351) compared with neutralization of ancestral D614G (B.1) in serum samples collected 6 weeks (a, b), 3 months (a, b), and 6 months (a) after the first vaccine dose from 2× BNT162b2 vaccinated HCWs ($n = 52$). MNT for D614G, Alpha, and Delta was performed with 4-day incubation (a) and for D614G, Beta, and Eta with 3-day incubation (b). Half-maximal inhibitory dilutions ($ID_{50}$) were analyzed with live viruses in microneutralization test, and titers <20 were marked as 10. Bars indicate geometric mean titers with geometric SDs. Fold changes between neutralization of variants by 6-wk and 3-mo samples (a, b) and 6-wk and 6-mo samples (a) are indicated below the bars Statistical differences in virus neutralization were analyzed with Wilcoxon matched-pairs signed rank test using Pratt's method, and $P$ values < 0.05 were considered statistically significant. ***, <0.001; ****, <0.0001. (c) Neutralization titers for virus isolates used in microneutralization test were compared. All follow-up serum samples (6 wk, 3 mo, and 6 mo postvaccination) were included in the analysis. Correlation was assessed with two-sided Spearman correlation test, and $P$ values < 0.05 were considered significant. Linear regression is shown with 95% confidence intervals. r, Spearman correlation coefficient.

this study, summer and autumn 2021) was analyzed with microneutralization test (MNT). MNTs for Alpha and Delta were performed with 4-day incubation (Fig. 3a), while MNTs for Beta and Eta were performed with 3-day incubation (Fig. 3b). MNT for D614G was performed with both 3-day and 4-day incubations (Fig. 3a and b).

**FIG 2** Legend (Continued)
D614G (B.1), Alpha (B.1.1.7), Eta (B.1.525), Beta (B.1.351), and Delta (B.1.617.2) variants. (b) Positioning of amino acid changes in SARS-CoV-2 trimeric spike protein structure (PDB ID: 6VXX) for Alpha (B.1.1.7), Beta (B.1.351), and Delta (B.1.617.2) variants. Surface is shown for spike protein S1 subunit and amino acid substitutions in S1 and S2 are displayed with magenta and amino acid deletions with green.

**TABLE 1** Neutralization of SARS-CoV-2 variants by BNT162b2 vaccinated HCWs[a]

| Virus strain | Incubation time | Samples | GMT ± SD | 95% CI | Positive (%) |
|---|---|---|---|---|---|
| D614G (B.1) | 4 days | 6 wk | 441 ± 2 | 359–543 | 100.0 |
| | 3 d | 6 wk | 354 ± 2 | 288–435 | 100.0 |
| | 4 d | 3 mo | 229 ± 2 | 180–292 | 100.0 |
| | 3 d | 3 mo | 136 ± 2 | 112–164 | 100.0 |
| | 4 d | 6 mo | 114 ± 2 | 94–139 | 100.0 |
| Alpha (B.1.1.7) | 4 d | 6 wk | 288 ± 2 | 231–360 | 100.0 |
| | 4 d | 3 mo | 182 ± 2 | 146–225 | 100.0 |
| | 4 d | 6 mo | 93 ± 2 | 78–111 | 100.0 |
| Delta (B.1.617.2) | 4 d | 6 wk | 117 ± 2 | 96–143 | 98.1 |
| | 4 d | 3 mo | 64 ± 2 | 53–79 | 96.2 |
| | 4 d | 6 mo | 31 ± 2 | 26–37 | 84.6 |
| Eta (B.1.525) | 3 d | 6 wk | 242 ± 2 | 192–305 | 100.0 |
| | 3 d | 3 mo | 116 ± 2 | 96–140 | 100.0 |
| Beta (1.351) | 3 d | 6 wk | 61 ± 2 | 50–76 | 96.2 |
| | 3 d | 3 mo | 31 ± 2 | 26–38 | 84.6 |

[a]GMT, geometric mean titer; 95% CI, 95% confidence interval of geometric mean. Neutralization titers were determined with microneutralization test using 3-day and 4-day incubation times for serum samples collected 6 weeks, 3 months, and 6 months after the first BNT162b2 vaccine dose. All vaccinees were vaccinated twice with a 3-week dose interval. Neutralization titers >=20 were considered positive.

Six weeks after the first vaccine dose (and 3 weeks after the second dose), all vaccinated HCWs had high neutralization titers against D614G (GMT 354, CI 288–435 with 3-day incubation and GMT 441, CI 359–543 with 4-day incubation), Alpha (GMT 288, CI 231–360), and Eta (GMT 242, CI 192–305), while the neutralization titers against Beta (GMT 61, CI 50–76) and Delta (GMT 117, CI 96–143) were reduced 5.8-fold and 3.8-fold compared to B.1, respectively (Fig. 3a and b, Table 1). Regardless of the reduction, Beta was neutralized by 96.2% (50/52) and Delta by 98.1% (51/52) of vaccinees' sera (Table 1).

Three months after vaccination, neutralizing antibody titers declined and the neutralization of D614G, Alpha, Eta, Beta, and Delta was reduced by 1.9–2.6-, 1.6-, 2.1-, 2.0-, and 1.8-fold, respectively, compared to the titers at 6 weeks (Fig. 3a and b). At 3 months after the vaccination, all vaccinees still had neutralizing antibodies against D614G, Alpha, and Eta variants, while Beta was neutralized by 84.6% (44/52) and Delta by 96.2% (50/52) of vaccinees' sera (Table 1).

Since Beta was replaced by Delta variant at the end of June 2021 and Eta was subsequently reclassified from VOI to VBM, we continued to monitor the neuralization efficacy of sera against the ancestral D614G variant and the circulating variants in Finland, Alpha and Delta. Six months postvaccination the fold-reduction of neutralizing antibody titers was increased to 3.9, 3.1, and 3.8 against D614G, Alpha, and Delta, respectively (Fig. 3a). Despite the decrease in the levels of neutralizing antibodies, sera of all vaccinees continued to neutralize D614G and Alpha, and 84.6% (44/52) neutralized Delta variant 6 months postvaccination (Table 1). Neutralization titers against Alpha (B.1.1.7) and Delta (B.1.617.2) variants had a strong correlation with each other ($r = 0.753$, $P < 0.0001$) and even stronger correlation with ancestral D614G strain ($r = 0.786$ and $r = 0.861$, respectively, $P < 0.0001$) (Fig. 3c). In addition, neutralization titers against Eta (B.1.525) and Beta (B.1.351) correlated with each other ($r = 0.764$, $P < 0.0001$) and moderately with D614G ($r = 0.611$ for Eta and $r = 0.508$ for Beta, $P < 0.0001$). However, high neutralization titers against D614G, Alpha and Eta were not necessarily associated with high titers against Delta and Beta variants (Fig. 3c).

Vaccinees were grouped with moderate ($ID_{50}$ 60–320) and high ($ID_{50}$ >320) neutralization titers, and the fold-reduction for D614G, Alpha, and Delta (6 wk versus 6 mo) was calculated. For D614G the fold-reduction was 2.8 in the moderate and 5.4 in the high titer group, whereas the fold-reductions were 2.5 and 5.1 for Alpha, and 4.0 and 11.8 for Delta (Fig. S1), indicating a steeper decline of neutralizing antibodies against Delta variant.

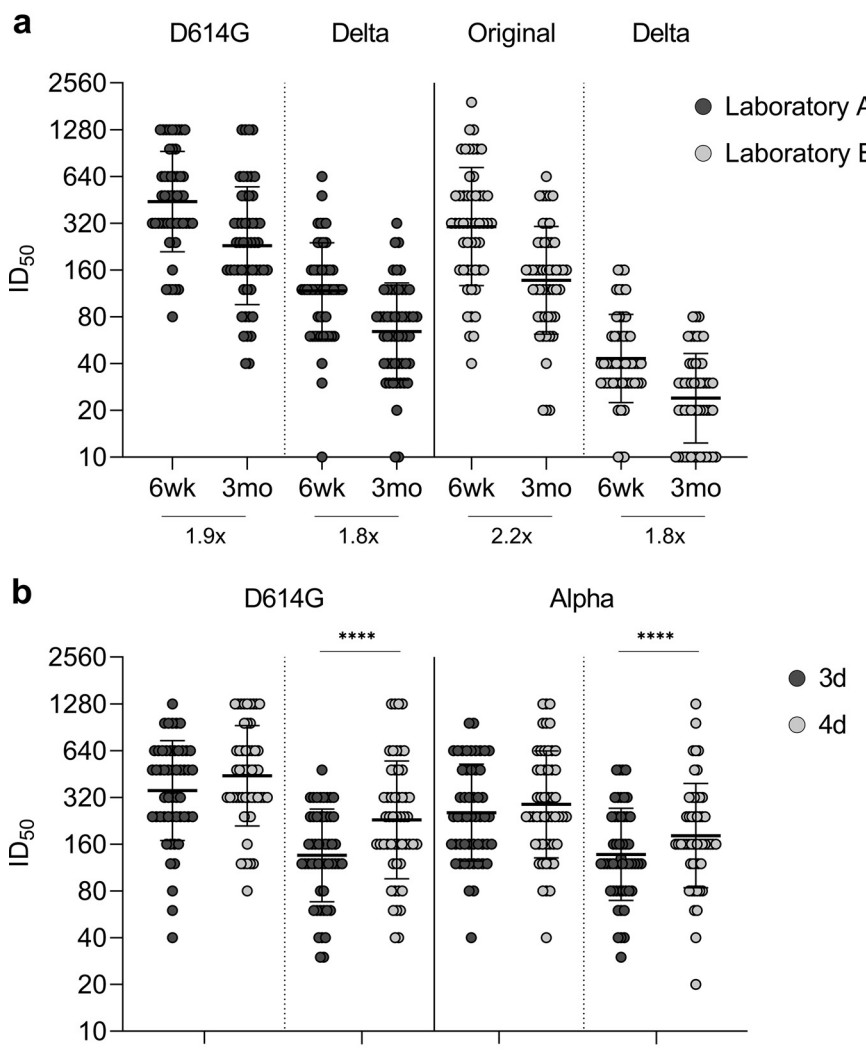

**FIG 4** Comparison of microneutralization tests. (a) Neutralization titers against D614G (B.1), original Wuhan-like variant (B), and Delta variant (B.1.617.2) were analyzed in microneutralization test performed in two laboratories (A and B) that use different cell lines (VeroE6-TMPRRS2 versus VeroE6), virus amounts (50 $TCID_{50}$ versus 100 $TCID_{50}$), incubation times (4 days versus 3 days), and reference virus strains (B.1 versus B). Fold changes between neutralization of variants by 6-wk and 3-mo samples are indicated below the bars. (b) Comparison of MNTs performed with 3- and 4-day incubation times in laboratory A for D614G and Alpha variants (VeroE6-TMPRRS2 cells and 50 $TCID_{50}$). Serum samples were collected 6 weeks and 3 months after the first vaccine dose from BNT162b2 vaccinated health care workers ($n = 52$). Neutralization titers $< 20$ were marked as 10. Bars indicate geometric mean titers (GMTs) with geometric SDs. Differences between incubation times were analyzed with Wilcoxon matched-pairs signed rank test using Pratt's method, and $P$ values $< 0.05$ were considered statistically significant. ****, $P < 0.0001$.

**Comparison of microneutralization tests.** Neutralization titers can be measured with several assay modifications impairing the comparisons of results between different reports. To confirm the neutralization titers, we performed microneutralization tests (MNTs) in two laboratories, A and B, which used different cell lines (VeroE6-TMPRRS2 versus VeroE6), virus dilutions (50 $TCID_{50}$ versus 100 $TCID_{50}$), and reference virus strains (B versus B.1), respectively. As shown in Fig. 4, despite different assay settings, sera from all vaccinees neutralized ancestral D614G (B.1) (laboratory A) and original Wuhan-like (B) (laboratory B) strains 6 weeks (GMT 441, CI 359–543, against D614G, and GMT 304, CI 239–388, against original strain) and 3 months (GMT 229, CI 180–292, against D614G and GMT 137, CI 110–172, against original strain) after vaccination (Fig. 4a). However, neutralization titers against Delta (B.1.617.2) differed between laboratories with GMTs 117 (CI

96–143) and 64 (CI 53–79) in laboratory A and 43 (CI 36–52) and 24 (CI 20–29) in laboratory B at 6 weeks and 3 months after vaccination, respectively. Despite different GMTs, fold-reduction in neutralization titers against Delta (6 wk versus 3 mo) was 1.8-fold in both laboratories. At 3 months postvaccination, neutralizing antibodies were detected against Delta in 96.2% and 73.1% of vaccinees at laboratory A and laboratory B, respectively.

In addition to comparison of MNTs performed in two laboratories, we tested 3- and 4-day incubation times for MNT in laboratory A for D614G and Alpha variants (Fig. 4b). GMTs were 354 (CI 288–435) and 441 (CI 359–543) against D614G, and 255 (CI 209–312) and 288 (CI 231–360) against Alpha (B.1.1.7) with 3- and 4-day incubation, respectively, indicating that longer incubation time increased neutralization titers slightly.

## DISCUSSION

Emerging SARS-CoV-2 variants can become dominant due to increased transmissibility and higher infectivity, and can also escape from neutralizing antibodies induced by infection and vaccinations (2). At the time of this study (summer and autumn 2021), the circulating dominant variant in Finland was Delta (B.1.617.2) (15), while a new variant, Omicron (B.1.1.529), has subsequently spread rapidly after its emergence at the end of November 2021 (2). Here, we demonstrate high levels of SARS-CoV-2 spike protein-binding and virus-neutralizing antibodies in the sera of BNT162b2 vaccinated HCWs after the second vaccine dose, and a slight decrease in antibody levels up to 6 months. Despite the decline of antibody levels, all vaccinated HCWs had neutralizing antibodies against original SARS-CoV-2 and Alpha variant at 6 months postvaccination. The Delta variant was neutralized less efficiently, but 6 months postvaccination, 84.6% of HCWs had neutralizing antibodies against Delta.

Previous studies have shown a decline in N- and S1-specific antibodies after SARS-CoV-2 infection (16, 17). Anti-S IgA and IgM antibodies have been shown to decay faster than IgG antibodies (16, 17); however, IgG and neutralizing antibodies seem to persist for at least a year after SARS-CoV-2 infection (18, 19). Similarly, a study among mRNA-1273 COVID-19 vaccinees showed that anti-RBD antibodies persist over 6 months despite a decrease in EIA titers (20). In addition, reduction of neutralization titers against SARS-CoV-2 variants in BNT162b2 vaccinees has been detected when followed up to 5 weeks after the second dose (21, 22). Our results extend these observations by showing that BNT162b2 vaccine-induced S1-specific and virus-neutralizing antibodies decline but remain detectable for months after the second vaccine dose. We confirmed our results in two laboratories using slightly different microneutralization test settings, and as the longer incubation time increased the titers slightly, the combination of differences in cell lines, amount of virus, and incubation times had the highest impact on the titers against Delta variant. However, the overall results were similar, showing the same fold reduction and the majority of vaccinees with neutralizing antibodies against Delta variant 6 months postvaccination.

Amino acid changes in spike proteins of variants contribute to immune evasion, and it has been suggested that N501Y is associated with increased infectivity (23), whereas L452R, T478K, and E484K with K417N reduce the interaction of neutralizing antibodies with RBD (24–26). SARS-CoV-2 variants in this study had variant-defined amino acid changes in the spike protein. We observed similar neutralization titers against D614G, Alpha, and Eta, whereas neutralization of Beta and Delta was reduced or lost 6 months postvaccination. Similar findings have been observed by others (26), and especially amino acid change E484K in Beta has been linked to immune evasion (9). However, Eta with slightly reduced neutralization results has solely an E484K substitution, indicating that the E484K substitution alone is not able to substantially decrease the neutralization efficacy of vaccine-induced antibodies. Importantly, regardless of the amino acid changes, 3 weeks after the second vaccine dose, 98.1% of vaccinees had neutralizing antibodies against Delta and 96.2% against Beta, indicating that the vaccine induces high levels of neutralizing antibodies against emerging

variants. Recently, results of a clinical trial with 6 months follow-up on BNT162b2 vaccinees showed 86–100% vaccine efficacy against COVID-19 disease from 7 days after the second dose to the end of the surveillance period (27). They also showed vaccine efficacy of 100% in South Africa at the time when Beta variant was prevalent. Our results on BNT162b2 vaccine-induced neutralizing antibodies are well in line with the vaccine efficacy results, showing high neutralizing capacity of serum antibodies during intervals of 6 weeks to 6 months postvaccination. The cell-mediated immunity contributes also to vaccine efficacy, and further analyses on T-cell immune responses in vaccinees are warranted.

Our study has some limitations: (i) The age distribution of the vaccinees does not represent the vaccinated population in Finland, and the results reflect antibody responses only among people of working age; (ii) HCWs are a selected socioeconomic group compared to the population of the entire country; and (iii) cell-mediated immunity was not included in this study.

The rapid spread of emerging variants, such as Omicron, highlights the importance of SARS-CoV-2 surveillance and vaccine efficacy and effectiveness studies. Our results indicate that the BNT162b2 vaccine induces high levels of antibodies that can neutralize emerging variants months after the vaccination. Further follow-up studies are warranted to determine the persistence of vaccine-induced antibodies, to reveal the role of cell-mediated immunity on vaccine efficacy, and to discover the neutralizing effect of vaccine-induced immunity on emerging variants to facilitate the decisions on additional vaccinations against SARS-CoV-2.

## MATERIALS AND METHODS

**Study participants.** HCWs ($n = 52$), who received two doses of BNT162b2 mRNA COVID-19 vaccine (BioNTech-Pfizer), encoding the S protein based on original SARS-CoV-2 isolate Wuhan Hu-1, at a 3-week interval at occupational health care, were selected from a larger cohort of vaccinated HCWs from Turku University Hospital (the Southwest Finland health district ethical permission ETMK 19/1801/2020 and EudraCT 2021–004419-14) and Helsinki University Hospital (the Helsinki-Uusimaa health district ethical permission HUS/1238/2020 and EudraCT 2021–004016-26)(8). HCWs were randomly selected from the cohort to obtain even gender distribution within the age range 22–65 years. Sera were collected before or on the day of the first vaccine, followed by 3-week, 6-week, 3-month, and 6-month sampling. Three-week sera were collected prior to the second vaccine dose. Written informed consent was obtained from all participants.

**SARS-CoV-2 S1- and N-protein based immunoassays.** Levels of SARS-CoV-2 S1 and N specific antibodies were analyzed with an in-house EIA (8, 28). Briefly, purified recombinant SARS-CoV-2 antigens based on Wuhan Hu-1 isolate were coated on 96-well plates (2.0 $\mu$g/mL of N and 3.5 $\mu$g/mL of S1). Serum samples were diluted 1:300 according to optimized protocol in the previous study (28). IgG and total Ig (IgG, IgA, and IgM) levels were determined with absorbance measurement at 450 nm wavelength. Optical density (OD) values were converted to EIA units using linear interpolation between OD-value of positive control (= 100 EIA units) and OD-value of negative control (= 0 EIA units). Thresholds to determine seropositivity were calculated separately for S1- and N-protein-based assays as the average of 30 randomly selected serum samples collected at Turku University Hospital before the COVID-19 pandemic plus three times the standard deviation. The samples used for threshold calculation have been described earlier (8).

**SARS-CoV-2 variants.** SARS-CoV-2 isolates: FIN1-20 (lineage B; GenBank accession number MZ934691 for passaged virus and GISAID accession number EPI_ISL_407079 for the original patient sample), FIN25-20 (B.1, D614G; MW717675.1 and EPI_ISL_412971), FIN32-21 (B.1.351, Beta variant; OK448476.1 and EPI_ISL_3471851), FIN33-21 (B.1.525, Eta variant; OK638135 and EPI_ISL_3471854), FIN35-21 (B.1.1.7, Alpha variant; OK448478.1 and EPI_ISL_2589882), and FIN37-21 (B.1.617.2, Delta variant; OK626882.1/Laboratory A, MZ945494/Laboratory B, and EPI_ISL_2557176). Viruses were isolated from nasopharyngeal samples by inoculating VeroE6 (FIN1-20 and FIN25-20) or VeroE6-TMPRRS2-H10 cells (29) (FIN32-21, FIN33-21, FIN35-21, and FIN37-21) and further passaged in VeroE6 cells (FIN1-20 and Fin37-21; Laboratory B) in Eagle's minimum essential medium (EMEM) supplemented with 2% fetal bovine serum (FBS), 2 mM L-glutamine, 20 mM HEPES, and penicillin-streptomycin or VeroE6-TMPRRS2-H10 cells (29) (FIN25-20, FIN32-21, FIN33-21, FIN35-21, and FIN37-21; Laboratory A) in DMEM supplemented with 2% FBS, 2 mM L-glutamine, and penicillin-streptomycin.

The SARS-CoV-2 isolates used in MNT (FIN1-20 passage 4 (p4), FIN25-20 p5, FIN32-21 p5, FIN33-21 p5, FIN35-21 p4, FIN37-21/Laboratory A p3, and FIN37-21/Laboratory B p3) were titered with TCID$_{50}$ assay. Briefly, in laboratory A (University of Turku) 50 000 VeroE6-TMPRRS2-H10 cells per well were seeded on flat-bottom 96-well plates on a previous day. Ten-fold dilutions of virus were added in DMEM supplemented with 2% FBS, 2 mM L-glutamine, and penicillin-streptomycin. After 3 days, cells were fixed with 10% formalin and stained with crystal violet. Laboratory B (Finnish Institute for Health and Welfare) TCID$_{50}$ assay for FIN1-20 and FIN37-21 viruses was done with $1/2$ log$_{10}$ virus dilutions in EMEM supplemented with 2% FBS and penicillin-streptomycin. Plates were incubated for 1 h at +37°C, and VeroE6

cells were added and incubated at +37°C, 5% CO2, for 4 days. Cells were fixed with 30% formalin and stained with crystal violet. $TCID_{50}$ was calculated with the Reed-Muench method, and the titers varied from $3 \times 10^5$ to $5 \times 10^7$ $TCID_{50}$/mL.

Full genome sequences of virus isolates used in MNT were obtained as described earlier (8) with IDT ARTIC nCoV-2019 V3 Panel. Sequence reads were mapped to the reference genome with BWA-MEM, and amino acid changes were identified with Nextclade (https://clades.nextstrain.org/).

Of note, passaging in VeroE6 cells cause mutations on the furin cleavage site on S/S2 that could affect MNT results. However, we did not notice any significant difference between neutralization titers against Alpha isolate used in this study (one substitution, R682W, in the furin cleavage site in all the sequence reads) and Alpha isolate used in our previous study (8) (intact furin cleavage site).

**SARS-CoV-2 microneutralization test.** To compare methods and verify results, MNTs were carried out in two laboratories. The two MNTs were essentially similar with slight modifications.

Laboratory A performed MNTs with slight modifications to the previously published method (8). Briefly, on flat-bottom 96-well plates, 2-fold serum dilutions (50 $\mu$l per well, in DMEM supplemented with 2% FBS, 2 mM L-glutamine, penicillin-streptomycin) were incubated with 50 $TCID_{50}$ virus per well for 1 h at +37°C. VeroE6-TMPRSS2-H10 cells were added (50,000 cells per well in 50 $\mu$l of the same medium) and incubated at +37°C, 5% $CO_2$, for 3 or 4 days. To visualize cell death, cells were fixed with 10% formalin and stained with crystal violet.

Laboratory B carried out MNTs as previously described (30). Briefly, on flat-bottom 96-well plates, 2-fold serum dilutions (50 $\mu$l per well, in EMEM supplemented with 2% FBS and penicillin-streptomycin) were incubated with 100 $TCID_{50}$ virus per well for 1 h at +37°C. VeroE6 cells were added (50,000 cells per well in 100 $\mu$l of the same medium) and incubated at +37°C, 5% CO2, for 4 days. Cells were fixed with 30% formalin and stained with crystal violet.

The first serum dilution was 1:20. The neutralizing titer was determined as the reciprocal of serum dilution inhibiting 50% of cell death. Neutralization titer of 1:20 or above was considered positive.

**Statistical analysis.** All serum samples were tested in duplicates. Statistical analysis was performed with GraphPad Prism 8. Statistically significant differences were tested with Wilcoxon matched-pairs signed-rank test with Pratt's method for handling of ties, and two-tailed $P$ values $< 0.05$ were considered statistically significant. A positive association between neutralization titers was analyzed with Spearman correlation. Amino acid changes were illustrated on SARS-CoV-2 spike protein structure (PDB: 6VXX) with UCSF Chimera 1.15.

**Data availability.** Original sequence data have been deposited in GenBank under accession numbers MZ934691, MW717675.1, OK448476.1, OK638135, OK448478.1, OK626882.1, and MZ945494. The raw data supporting the conclusions of this article are available upon request from the corresponding authors.

## SUPPLEMENTAL MATERIAL

Supplemental material is available online only.
**SUPPLEMENTAL FILE 1,** PDF file, 0.2 MB.

## ACKNOWLEDGMENTS

We thank Sari Pakkanen, Outi Debnam, Simo Miettinen, Anne Suominen, Anne-Mari Pieniniemi, Marja-Liisa Ollonen, Tiina Sihvonen, and Johanna Rintamäki for technical assistance. All the volunteer HCWs are thanked for their commitment to donate follow-up serum specimens. We gratefully acknowledge originating and submitting laboratories for genetic data in the GISAID EpiCoV database.

This work was supported by the Jane and Aatos Erkko Foundation (grant numbers 3067-84b53 and 5360-cc2fc to I.J.); the Academy of Finland (grant numbers 336410 and 337530 to I.J., and 336439 and 335527 to A.K.); the Sigrid Jusélius Foundation (to I.J.); the Finnish Medical Foundation (to A.K.); and The Turku University Hospital Research Foundation (to P.A.T., L.I., and J.L.). This work did not receive funding from any vaccine manufacturer.

P.J., A.H., A.K., I.J., J.L., and L.K. designed the experiments; P.J., P.K., A.H., M.H., L.L., P.Ö., S.M., A.R., M.B., T.S., and L.K. performed the experiments; P.A.T., L.I., H.K.H., E.O., A.K., and J.L. contributed to data collection and data design; P.J., P.K., A.H., T.S., and L.K. analyzed the data; P.J., I.J., and L.K. wrote the manuscript; and all authors revised and approved the manuscript for publication.

We declare no competing interests.

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
