## [Reviewer comments · Microbiology Spectrum]

Microbiology Spectrum

Vaccine-induced antibody responses against SARS-CoV-2 variants-of-concern six months after the BNT162b2 COVID-19 mRNA vaccination

Pinja Jalkanen, Pekka Kolehmainen, Anu Haveri, Moona Huttunen, Larissa Laine, Pamela Österlund, Paula Tähtinen, Lauri Ivaska, Sari Maljanen, Arttu Reinholm, Milja Belik, Teemu Smura, Hanni Häkkinen, Eeva Ortamo, Anu Kantele, Ilkka Julkunen, Johanna Lempainen, and Laura Kakkola

Corresponding Author(s): Pinja Jalkanen, University of Turku

Review Timeline:

Submission Date:	November 12, 2021
Editorial Decision:	January 4, 2022
Revision Received:	January 12, 2022
Editorial Decision:	January 25, 2022
Revision Received:	January 28, 2022
Accepted:	February 6, 2022

Editor: Mathilde Richard

Reviewer(s): Disclosure of reviewer identity is with reference to reviewer comments included in decision letter(s). The following individuals involved in review of your submission have agreed to reveal their identity: Rory D. de Vries (Reviewer #1)

Transaction Report:

DOI: <https://doi.org/10.1128/spectrum.02252-21>

January 4, 2022

Dr. Pinja Jalkanen
University of Turku
Institute of Biomedicine
Kiinamylynkatu 10
Turku 20520
Finland

Re: Spectrum02252-21 (Vaccine-induced antibody responses against SARS-CoV-2 variants-of-concern six months after the COVID-19 mRNA vaccination)

Dear Dr. Pinja Jalkanen:

Link Not Available

Sincerely,

Mathilde Richard

Journals Department
Reviewer comments:

Reviewer #1 (Comments for the Author):

In this manuscript, Jalkanen et al describe BTN162b2 COVID-19 vaccine-induced neutralizing antibody responses against different SARS-CoV-2 variants. They find a gradual decrease in antibodies between 3 weeks after second vaccination and 6 months after vaccination, and also describe reduced neutralizing antibody titers against Delta. The authors have presented the data well; however, I have some comments the authors should address:

Major comments:

- The main message of the manuscript is not new. Longevity of immune responses to different variants has already been reported in literature. However, the authors have an elegant description, with a geographically distinct population, so this is not a dealbreaker (especially not for a journal that seeks to publish technically sound studies regardless of impact).
- The authors mention that this manuscript is about BNT162b2 for the first time in line 75. This should also be mentioned in the title, abstract and importance paragraphs. Generalization to 'COVID-19 mRNA vaccines' is not clear, and the authors should be

specific. Same goes for results (e.g. line 94).

- Line 101-102: The authors state that all vaccinees had S1-specific IgG 6 months after vaccination. What is the responder cut-off in this in-house developed ELISA, and how was this cut-off validated? Concretely: how are the authors able to define a responder?

- I find the results section a bit confusing, and the authors should structure this better. E.g., in line 122-123 the authors use a header on neutralization of B.1.525 and B.1.351, but most of the paragraph discusses B.1.617.2.

- The authors have performed MNT at two different sites (which is great), but find a site-specific difference. This should be discussed in the discussion paragraph.

- In line 177-178 (first line of the discussion), the authors state that variants can become dominant due to 'escape from neutralizing antibodies'. Can the authors back this up with a reference, proving that immune escape leads to dominance?

Methodological comments:

- The authors have performed ELISAs with a single serum dilution (1:300). Were titrations performed to determine that this is indeed the optimal dilution?

- Characterization of viruses is crucial in infectious virus neutralization assays. The authors isolated and passaged the different viruses on VeroE6, which we know can cause laboratory adaptations. Two questions:

o Which passage was sequenced? Are the authors aware of the potential for adaptation in VeroE6?

o Did the authors assess infectivity of VeroE6 cells for the different variants? Is it possible that adaptation of the stocks to VeroE6 influences the results of the MNT assay?

Minor comments:

- The authors use the terms Beta and Delta in the abstract, but B.1.351 and B.1.617.2 in the rest of the manuscript. To be consistent, and adhere to WHO recommendations, I suggest that the authors stick to the Greek alphabet naming for variants if possible.

- Line 51-53: statements like these are so difficult to keep up-to-date. Omikron is now classified as VOC, consider updating the introduction.

- Line 63: the authors should use 'correlated', rather than 'associated'.

- Line 84: the authors should use 'in', rather than 'on'.

- Figure 4. It would be better to show 2-log transformed data for these correlations.

Reviewer #2 (Comments for the Author):

Jalkanen et al., have performed an analysis of the antibody response in a cohort of vaccinated healthcare workers (HCW) in Finland. HCW's received the BNT162b2 mRNA COVID vaccine and the antibody response was monitored out to 6 months. The ability of serum samples to neutralize an isolate closely related to the vaccine strain and to cross neutralize several variants of concern (VOC) was analyzed. The authors found the neutralizing antibody titers against the vaccine strain declined over 6 months; however, despite this reduction, all vaccinees maintained an antibody response for at least 6 months. Neutralizing titers were also reduced against different VOCs over the 6-month period, and the VOC with the greatest reduction in antibody neutralization was B.1.617.2. The authors extend the findings of other groups, and show that despite a reduction in neutralizing titers, >80% of patients retained the ability to neutralize the B.1.617.2 variant at 6 months post-vaccination.

Major comments:

1) The manuscript is scientifically sound, but it is very difficult to follow and needs to be edited for clarity. Specifically, Fig S1 is referenced heavily and needs to be included in the main text. If possible, Fig S2 should also be included in the main text. Fig 1 should be mentioned in the introduction lines 71-80. Line 89-90 reads: four samples were collected 2, 2, 7, and 12 days after the first vaccine dose. It is unclear what is meant by this statement. Prior to line 207 the manuscript is about the antibody response, and then the manuscript suddenly pivots and starts to discuss sequencing of viruses. There needs to be a transition explaining why the authors are now sequencing viruses, the source of the viruses, and the greater role in the paper. The role of these viruses does not become clear until reading further into the manuscript.

In line 122, the authors begin to discuss their results on antibody analysis and this section addresses the response at 3 months post-vaccination. The authors first discuss the response to the B.1 and B strains and then in lines 142-149, discuss the response to the B.1.525 and B.1.351 variants. In the following section, starting with line 152, the authors then re-explain the response to the B.1 and B strains, and this is very confusing and difficult to follow. It would greatly improve the manuscript to discuss the antibody response at 3 weeks, 6 weeks, 3 months, and 6 months to the B, B.1, B.1.1.7 and B.1.617.2 variants in a single section and then, after discussing this response, in a separate section discuss the response to the B.1.525 and B.1.351 variants.

2) The serum samples were analyzed by microneutralization assay in two different laboratories and the results were compared. The authors mention in the main text that Laboratory A used a 3 day incubation while Laboratory B used a 4 day incubation. They explain this resulted in subtle differences in sensitivity. However, in the methods section, Laboratory A used 50 TCID50 while Laboratory B used 100 TCID50 in their neutralization assays. This would also contribute to variability in the results, and it

is likely that the assay in laboratory A would generate higher neutralization titers. The authors need to mention and discuss this difference in methods in the main text. This is particularly relevant given the authors have chosen to emphasize the findings in Fig 3 from Laboratory A which used 50 TCID₅₀ in the neutralization assay and this may have resulted in elevated titers.

Minor comments:

- 1) Line 136. Please indicate the described results are shown in Fig 3
- 2) Lines 139-141. The comparison be made is difficult to understand. Please clearly indicate which data in Fig 3 is being compared to the appropriate panel in Fig S1.
- 3) Lines 152-155. Please clearly indicate the data is being presented in Fig 3.
- 4) Line 161-162. Please include the data on fold reductions between 6 weeks and 3 months in Fig 3 or on Table 1. The fold reduction data is only shown in the text but not on any of the figures.
- 5) Lines 169-170. Please modify. The figure shows a correlation analysis with the regression displayed by 95% CI in grey.
- 6) Lines 212-216. The author's reference a study on vaccine-efficacy. It is important to note that vaccine-efficacy is measured by a reduction in disease severity. The authors should discuss the antibody response described in the vaccine study and how it relates to the current findings.
- 7) Figure 1 and Figure 1 legend. Please clarify Line 484-485. From review of this sentence, it is unclear if the numerical value shown above each bar on the graph is the SD or the GM. It is assumed to be the GM, but please clarify in the figure legend.

Staff Comments:

Preparing Revision Guidelines

Please return the manuscript within 60 days; if you cannot complete the modification within this time period, please contact me. If you do not wish to modify the manuscript and prefer to submit it to another journal, please notify me of your decision immediately so that the manuscript may be formally withdrawn from consideration by Microbiology Spectrum.

Re: Spectrum02252-21 (Vaccine-induced antibody responses against SARS-CoV-2 variants-of-concern six months after the COVID-19 mRNA vaccination)

We thank the Editor and the Reviewers for their careful evaluation of our manuscript and their comments to further improve it. Please find below our responses to all concerns and comments raised by the reviewers.

Reviewer #1 (Comments for the Author):

In this manuscript, Jalkanen et al describe BTN162b2 COVID-19 vaccine-induced neutralizing antibody responses against different SARS-CoV-2 variants. They find a gradual decrease in antibodies between 3 weeks after second vaccination and 6 months after vaccination, and also describe reduced neutralizing antibody titers against Delta. The authors have presented the data well; however, I have some comments the authors should address:

We thank Reviewer #1 for excellent comments and scientific views that greatly improved our manuscript. We have now responded to all the comments and edited our manuscript accordingly.

Major comments:

- The main message of the manuscript is not new. Longevity of immune responses to different variants has already been reported in literature. However, the authors have an elegant description, with a geographically distinct population, so this is not a dealbreaker (especially not for a journal that seeks to publish technically sound studies regardless of impact).

We agree that others have also published similar data (referenced in our manuscript). However, as the Reviewer states, our study focuses on geographically distinct population and further expands the knowledge on decreasing vaccine-induced antibody levels as well as emphasizes the need to constantly discuss vaccine strategies. Importantly, our study uses live viruses isolated from local patients (i.e. locally circulating isolates), as also the less-studied VBM Eta (B.1.525) with interesting amino acid changes, thus adding more relevance to the study.

- The authors mention that this manuscript is about BNT162b2 for the first time in line 75. This should also be mentioned in the title, abstract and importance paragraphs. Generalization to 'COVID-19 mRNA vaccines' is not clear, and the authors should be specific. Same goes for results (e.g. line 94).

We have clarified this by adding BNT162b2 into title, other mentioned paragraphs, and additional relevant places in the text.

- Line 101-102: The authors state that all vaccinees had S1-specific IgG 6 months after vaccination. What is the responder cut-off in this in-house developed ELISA, and how was this cut-off validated? Concretely: how are the authors able to define a responder?

The in-house ELISA used in this study is described in detail in the reference number 8 (Jalkanen et al., Nat Commun, 2021) and there in publication by Jalkanen et al., JID, 2021 (<https://academic.oup.com/jid/article/224/2/218/6255671>). We have now, in addition to reference 8, included the reference for the JID publication in the methods section (reference nro 28, line 300 in Marked-Up Manuscript). For this study, the cut-off values were calculated as the average of 30 randomly selected serum samples collected at Turku University Hospital before the COVID-19 pandemic plus three times the standard deviation. The cut-off values were established separately for S1 and N-protein-based assays measuring anti-human IgG and total Ig (IgG + IgA + IgM). The resulting cut-off values were 9.7 and 16.0 EIA units for anti-S1 IgG and total Ig, and 7.7 and 8.3 EIA units for

anti-N IgG and total Ig, respectively. We did not use the same cut-off values as in the previous study (reference 8) since we had to change the COVID-19 patient samples used in the positive control pool due to the shortage of the previously used serum samples. However, both the previously used and the currently used positive control pools gave similar results and thus the cut-off values are comparable.

- I find the results section a bit confusing, and the authors should structure this better. E.g., in line 122-123 the authors use a header on neutralization of B.1.525 and B.1.351, but most of the paragraph discusses B.1.617.2.

We have now clarified the Results section by reorganizing the results. We also took into consideration the comments by Reviewer #2 raising similar issues. In our opinion the Results section is now clear and easy to follow.

The following major reorganizations were made:

- 1) Fig. 3 is now Fig. 3a
- 2) MNT results for B.1 (D614G), B.1.525 (Eta) and B.1.351 (Beta) from Supplementary Fig. 1a are now Fig. 3b
- 3) Fig. 4 is now Fig. 3c (same data represented in a bit different style) and includes additional data to compare D614G, Beta, and Eta
- 4) Data from Supplementary Fig. 1a for B.1 (D614G) and B.1.1.7 (Delta) and from Supplementary Fig. 1b for B (Original) and B.1.617.2 (Delta) are now included as Fig. 4a
- 5) Fig. 4b is a new figure with comparison of the effect of incubation time (3 or 4 days) on MNTs
- 6) Supplementary Fig. 2 is now Supplementary Fig. 1

- The authors have performed MNT at two different sites (which is great), but find a site-specific difference. This should be discussed in the discussion paragraph.

Due to reorganization of the results, we have now made a separate paragraph of this comparison to the Results section and separated the results now into a new Fig. 4. We have also included a short discussion of these results onto Discussion (lines 250-255 in Marked-Up Manuscript).

- In line 177-178 (first line of the discussion), the authors state that variants can become dominant due to 'escape from neutralizing antibodies'. Can the authors back this up with a reference, proving that immune escape leads to dominance?

We apologize this misleading phrasing and have now rewritten the sentence in lines 229-231 in Marked-Up Manuscript "Emerging SARS-CoV-2 variants can become dominant due to increased transmissibility and, higher infectivity, and can also escape from neutralizing antibodies induced by infection and vaccinations."

Methodological comments:

- The authors have performed ELISAs with a single serum dilution (1:300). Were titrations performed to determine that this is indeed the optimal dilution?

Optimal dilution was determined in our previous paper (reference 28 in resubmitted manuscript file). Different serum dilutions (between 1:100 to 1:10 000) were tested and the optimal dilution of 1:300 with high absorbance from positive COVID-19 serum samples and low background from negative serum samples was selected.

- Characterization of viruses is crucial in infectious virus neutralization assays. The authors isolated and passaged the different viruses on VeroE6, which we know can cause laboratory adaptations. Two questions:

o Which passage was sequenced? Are the authors aware of the potential for adaptation in VeroE6?

All SARS-CoV-2 isolates were passaged in either VeroE6 or VeroE6-TMPRSS2-H10 cell lines or in both cell lines as described in the manuscript. The passage used in the neutralization tests was sequenced: FIN1-20 passage 4 (p4), FIN25-20 p5, FIN32-21 p5, FIN33-21 p5, FIN35-21 p4, FIN37-21/Laboratory A p3, and FIN37-21/Laboratory B p3.

In our manuscript we have marked in Fig. 2 the amino acid deletions and substitutions (compared to original Wuhan-like sequence) that are present in over 20% of the NGS-obtained sequences on S-protein and that have accumulated upon propagation in the VeroE6 or VeroE6-TMPRSS2-H10 cells. We are aware of the cell culture adaptations and the impairing effect of these mutations on the cleavage of spike protein into S1 and S2 subunits (<https://elifesciences.org/articles/66815>).

o Did the authors assess infectivity of VeroE6 cells for the different variants? Is it possible that adaptation of the stocks to VeroE6 influences the results of the MNT assay?

Infectivity of different variants was not assessed directly. The amount of infectious virus in propagated virus stocks was determined with end point titration. All SARS-CoV-2 isolates used in the study had median tissue culture infectious dose (TCID₅₀) values that were in the same range (3x10⁵-5x10⁷) indicating a similar infection capability. It is possible that amino acid changes caused by adaptation into the VeroE6 cells could have some minor effect on MNT. However, we did not notice any significant difference between neutralization titers against Alpha isolate used in this study (one substitution, R682W, in the furin cleavage site in all the sequence reads) and Alpha isolate used in our previous results (reference 8, intact furin cleavage site). In addition, FIN25-20 (D614G) and FIN37-21 (Delta) passaged only in VeroE6-TMPRSS2-H10 cells in this study in Lab A did not have major (>50%) changes in their genome, indicating that majority of the sequence variants have retained their ability of S1/S2 cleavage.

Minor comments:

- The authors use the terms Beta and Delta in the abstract, but B.1.351 and B.1.617.2 in the rest of the manuscript. To be consistent, and adhere to WHO recommendations, I suggest that the authors stick to the Greek alphabet naming for variants if possible.

We have changed the nomenclature of SARS-CoV-2 variants from Pango lineage to WHO classification. However, we have included Pango lineage numbers in parenthesis in relevant places since Pango lineage numbers are also widely recognized by researchers and health agencies.

- Line 51-53: statements like these are so difficult to keep up-to-date. Omikron is now classified as VOC, consider updating the introduction.

We thank the Reviewer for acknowledging that it is very difficult to keep up with the speed of virus evolution. Indeed, we submitted this manuscript before Omicron begun to spread, but we have now mentioned Omicron in relevant parts of the Discussion section (lines 233 and 279 in Marked-Up Manuscript). We would like to keep Omicron out from the Introduction to avoid any confusion as to selection of VOCs at the time this study was done.

- Line 63: the authors should use 'correlated', rather than 'associated'.

We have now changed the text as suggested by the Reviewer.

- Line 84: the authors should use 'in', rather than 'on'.

We have now changed the text as suggested by the Reviewer.

- Figure 4. It would be better to show 2-log transformed data for these correlations.

We have now changed the Figure 3c (previously Figure 4) to 2-log scale.

Reviewer #2 (Comments for the Author):

Jalkanen et al., have performed an analysis of the antibody response in a cohort of vaccinated healthcare workers (HCW) in Finland. HCW's received the BNT162b2 mRNA COVID vaccine and the antibody response was monitored out to 6 months. The ability of serum samples to neutralize an isolate closely related to the vaccine strain and to cross neutralize several variants of concern (VOC) was analyzed. The authors found the neutralizing antibody titers against the vaccine strain declined over 6 months; however, despite this reduction, all vaccinees maintained an antibody response for at least 6 months. Neutralizing titers were also reduced against different VOCs over the 6-month period, and the VOC with the greatest reduction in antibody neutralization was B.1.617.2. The authors extend the findings of other groups, and show that despite a reduction in neutralizing titers, >80% of patients retained the ability to neutralize the B.1.617.2 variant at 6 months post-vaccination.

We thank the Reviewer 2 for the scientific view and comments to further improve our manuscript. We have now responded to all the comments and edited our manuscript accordingly.

Major comments:

1) The manuscript is scientifically sound, but it is very difficult to follow and needs to be edited for clarity. Specifically, Fig S1 is referenced heavily and needs to be included in the main text. If possible, Fig S2 should also be included in the main text. Fig 1 should be mentioned in the introduction lines 71-80. Line 89-90 reads: four samples were collected 2, 2, 7, and 12 days after the first vaccine dose. It is unclear what is meant by this statement. Prior to line 207 the manuscript is about the antibody response, and then the manuscript suddenly pivots and starts to discuss sequencing of viruses. There needs to be a transition explaining why the authors are now sequencing viruses, the source of the viruses, and the greater role in the paper. The role of these viruses does not become clear until reading further into the manuscript.

As also Reviewer #1 asked to clarify especially the Results section, we have now reorganized the Results part of the manuscript. The following major reorganizations were made (as described also above):

1) Fig. 3 is now Fig. 3a

2) MNT results for B.1 (D614G), B.1.525 (Eta) and B.1.351 (Beta) from Supplementary Fig. 1a are now Fig. 3b

3) Fig. 4 is now Fig. 3c (same data represented in a bit different style) and includes additional data to compare D614G, Beta, and Eta

4) Data from Supplementary Fig. 1a for B.1 (D614G) and B.1.1.7 (Delta) and from Supplementary Fig. 1b for B (Original) and B.1.617.2 (Delta) are now included as Fig. 4a

5) Fig. 4b is a new figure with comparison of the effect of incubation time (3 or 4 days) on MNTs

6) Supplementary Fig. 2 is now Supplementary Fig. 1

However, we feel that mentioning our results (Fig 1) in the Introduction is not relevant for the clarity of the manuscript. We feel that the changes suggested by the Reviewer #2, and now implemented, have significantly improved the clarity of our manuscript.

Line 89-90 (line 90 in Marked-Up Manuscript) is now corrected: 0-12 days

To clarify the role of the viruses, we have now added the sentence: “To determine the neutralization capacity of the vaccine-induced antibodies in microneutralization test with live viruses, SARS-CoV-2 variants representing five variants were isolated from Finnish COVID-19 patients.” to the beginning of the chapter (lines 109-111 in Marked-Up Manuscript). In addition, we have in the subsequent sentence clarified that the viruses were propagated in cells (line 114 in Marked-Up Manuscript).

In line 122, the authors begin to discuss their results on antibody analysis and this section addresses the response at 3 months post-vaccination. The authors first discuss the response to the B.1 and B strains and then in lines 142-149, discuss the response the B.1.525 and B.1.351 variants. In the following section, starting with line 152, the authors then re-explain the response to the B.1 and B strains, and this is very confusing and difficult to follow. It would greatly improve the manuscript to discuss the antibody response at 3 weeks, 6 weeks, 3 months, and 6 months to the B, B.1, B.1.1.7 and B.1.617.2 variants in a single section and then, after discussing this response, in a separate section discuss the response to the B.1.525 and B.1.351 variants.

We apologize the confusing presentation of our results. We have now reorganized the Results section and as suggested by the Reviewer, also reorganized the neutralizing antibody analysis results according to time. We foresee these changes have now greatly clarified the manuscript.

2) The serum samples were analyzed by microneutralization assay in two different laboratories and the results were compared. The authors mention in the main text that Laboratory A used a 3 day incubation while Laboratory B used a 4 day incubation. They explain this resulted in subtle differences in sensitivity. However, in the methods section, Laboratory A used 50 TCID₅₀ while Laboratory B used 100 TCID₅₀ in their neutralization assays. This would also contribute to variability in the results, and it is likely that the assay in laboratory A would generate higher neutralization titers. The authors need to mention and discuss this difference in methods in the main text. This is particularly relevant given the authors have chosen to emphasize the findings in Fig 3 from Laboratory A which used 50 TCID₅₀ in the neutralization assay and this may have resulted in elevated titers.

Due to reorganization of the results, we have now made a separate paragraph of this comparison to the Results section and separated the results now into a new Fig 4. We have also included a short discussion of these results into the Discussion section (lines 250-255 in Marked-Up Manuscript).

Minor comments:

1) Line 136. Please indicate the described results are shown in Fig 3

Due to reorganization of the results, this result is now represented in Figure 4b to which this part of the results is also indicated.

2) Lines 139-141. The comparison be made is difficult to understand. Please clearly indicate which data in Fig 3 is being compared to the appropriate panel in Fig S1.

Due to reorganization of the results, this result is now represented in Figure 3a-b and the compared data points are indicated more clearly.

3) Lines 152-155. Please clearly indicate the data is being presented in Fig 3.

Due to reorganization of the results, the data is now more clearly indicated in Figure 3 to which this part of the results is also indicated.

4) Line 161-162. Please include the data on fold reductions between 6 weeks and 3 months in Fig 3 or on Table 1. The fold reduction data is only shown in the text but not on any of the figures.

Requested fold reductions have now been added into Figure 3.

5) Lines 169-170. Please modify. The figure shows a correlation analysis with the regression displayed by 95% CI in grey.

We have now modified this sentence into “However, high neutralization titers against D614G, Alpha and Eta were not necessarily associated with high titers against Delta and Beta variants (Fig. 3c)” on lines 195-197 in Marked-Up Manuscript.

6) Lines 212-216. The author's reference a study on vaccine-efficacy. It is important to note that vaccine-efficacy is measured by a reduction in disease severity. The authors should discuss the antibody response described in the vaccine study and how it relates to the current findings.

We have now rewritten the sentence to emphasize the relevance of neutralizing antibodies to vaccine efficacy (lines 269-274 in Marked-Up Manuscript): “Recently, results of a clinical trial with 6 months follow-up on BNT162b2 vaccinees showed 96.7% vaccine efficacy against severe form of COVID-19 disease despite a slight decline in the overall vaccine efficacy against SARS-CoV-2 infection(27). This vaccine efficacy, measuring the effect of vaccine on the disease, results from vaccine-induced neutralizing antibodies, although cell-mediated immunity also has an important role.”

7) Figure 1 and Figure 1 legend. Please clarify Line 484-485. From review of this sentence, it is unclear if the numerical value shown above each bar on the graph is the SD or the GM. It is assumed to be the GM, but please clarify in the figure legend.

We have now modified the Figure 1 legend to clarify that values above each bar are GMs.

January 25, 2022

Dr. Pinja Jalkanen
University of Turku
Institute of Biomedicine
Kiinamylynkatu 10
Turku 20520
Finland

Re: Spectrum02252-21R1 (Vaccine-induced antibody responses against SARS-CoV-2 variants-of-concern six months after the BNT162b2 COVID-19 mRNA vaccination)

Dear Dr. Pinja Jalkanen:

Thank you for submitting your manuscript to Microbiology Spectrum. As you will see your paper is very close to acceptance. Your modified manuscript and rebuttal letter was sent back to Reviewer 1 who had concerns about the methodology. They judged that their comments were appropriately addressed (see below). However, there are some minor text modifications that I am requesting before acceptance. Please modify the manuscript along the lines I have recommended:

1. The methodology concerns raised by R1 were addressed in the rebuttal letter, but for clarity for the future readers, please add the provided information in the rebuttal letter to the Material and Methods.
2. For readers that are not very familiar with SARS-CoV-2, please indicate from which strain the spike mRNA present in the BNT162b2 vaccine is derived.
3. Line 90: change "0-12 days" to "up to 12 days after"
4. Fig 1: in the legend and in the main text referring to Figure 1, please indicate that the antibody levels were determined against the strain from which the spike mRNA present in the vaccine was derived.
5. Comment 6 of R2 was not addressed sufficiently. The findings from ref 27 should be put in perspective with the results from the current study.

As these revisions are quite minor, I expect that you should be able to turn in the revised paper in less than 30 days, if not sooner.

When submitting the revised version of your paper, please provide (1) point-by-point responses to the issues I raised in your cover letter, and (2) a PDF file that indicates the changes from the original submission (by highlighting or underlining the changes) as file type "Marked Up Manuscript - For Review Only". Please use this link to submit your revised manuscript. Detailed instructions on submitting your revised paper are below.

Link Not Available

Sincerely,

Mathilde Richard

Reviewer comments:

Reviewer #1 (Comments for the Author):

My suggestions were sufficiently addressed to warrant publication. Figure 4 could be a supplemental figure, as it focuses on assay validation, but it's also fine to leave it in as a main figure.

Preparing Revision Guidelines

- point-by-point responses to the issues I raised in your cover letter
- Upload a compare copy of the manuscript (without figures) as a "Marked-Up Manuscript" file.
- Each figure must be uploaded as a separate file, and any multipanel figures must be assembled into one file.
- Manuscript: A .DOC version of the revised manuscript
- Figures: Editable, high-resolution, individual figure files are required at revision, TIFF or EPS files are preferred

Please return the manuscript within 60 days; if you cannot complete the modification within this time period, please contact me. If you do not wish to modify the manuscript and prefer to submit it to another journal, please notify me of your decision immediately so that the manuscript may be formally withdrawn from consideration by Microbiology Spectrum.

Re: Spectrum02252-21R1 (Vaccine-induced antibody responses against SARS-CoV-2 variants-of-concern six months after the BNT162b2 COVID-19 mRNA vaccination)

Thank you for submitting your manuscript to Microbiology Spectrum. As you will see your paper is very close to acceptance. Your modified manuscript and rebuttal letter was sent back to Reviewer 1 who had concerns about the methodology. They judged that their comments were appropriately addressed (see below). However, there are some minor text modifications that I am requesting before acceptance. Please modify the manuscript along the lines I have recommended:

We thank the Reviewer and the Editor for these positive comments on our resubmitted manuscript. We have now introduced the required minor text modifications to the text and indicated those in detail below. We hope our manuscript is acceptable for publication after these modifications.

1. The methodology concerns raised by R1 were addressed in the rebuttal letter, but for clarity for the future readers, please add the provided information in the rebuttal letter to the Material and Methods.

We have now added the information about methodology from the rebuttal letter into the Material and Methods section. The following sentences were added/complemented:

on the EIA section lines 260-261 "Serum samples were diluted 1:300 according to optimized protocol in the previous study (28)." and lines 264-268 "Thresholds to determine seropositivity were calculated separately for S1 and N-protein-based assays as the average of 30 randomly selected serum samples collected at Turku University Hospital before the COVID-19 pandemic plus three times the standard deviation. The samples used for threshold calculation have been described earlier (8)";

on the SARS-CoV-2 variants section lines 284-286 "The SARS-CoV-2 isolates used in MNT (FIN1-20 passage 4 (p4), FIN25-20 p5, FIN32-21 p5, FIN33-21 p5, FIN35-21 p4, FIN37-21/Laboratory A p3, and FIN37-21/Laboratory B p3) were titrated with TCID₅₀ assay", lines 294-295 "TCID₅₀ was calculated with the Reed-Muench method and the titers varied from 3×10^5 to 5×10^7 TCID₅₀/ml", line 296 "Full genome sequences of virus isolates used in MNT were obtained as described earlier...", and lines 300-304 "Of note, passaging in VeroE6 cells cause mutations on the furin cleavage site on S1/S2 that could affect MNT results. However, we did not notice any significant difference between neutralization titers against Alpha isolate used in this study (one substitution, R682W, in the furin cleavage site in all the sequence reads) and Alpha isolate used in our previous study (8) (intact furin cleavage site)."

2. For readers that are not very familiar with SARS-CoV-2, please indicate from which strain the spike mRNA present in the BNT162b2 vaccine is derived.

To indicate the vaccine strain, we have modified the first sentence of the study participant part in the Methods section, lines 246-248: "HCWs (n=52), who received two doses of BNT162b2 mRNA Covid-19 vaccine (BioNTech-Pfizer), encoding the S protein based on original SARS-CoV-2 isolate Wuhan Hu-1, at a three-week interval at occupational healthcare...".

3. Line 90: change "0-12 days" to "up to 12 days after"

We have now changed the text as suggested, line 88.

4. Fig 1: in the legend and in the main text referring to Figure 1, please indicate that the antibody levels were determined against the strain from which the spike mRNA present in the vaccine was derived.

We have modified the results section on lines 92-95 as follows: “To analyze the longevity of IgG and total Ig antibodies after the BNT162b2 vaccination, the antibody levels were measured with enzyme immunoassay (EIA) against spike protein subunit S1 representing the original SARS-CoV-2 isolate Wuhan Hu-1 that is encoded also by the vaccine mRNA”.

We have now also mentioned in the figure legend that S1 and N proteins used in EIA represented Wuhan Hu-1 isolate, lines 520-521: “Fig. 1 Antibody responses in BNT162b2 vaccinated HCWs against SARS-CoV-2 spike glycoprotein subunit S1 and nucleoprotein (N) representing isolate Wuhan Hu-1”.

In addition, we have added the same information into the Methods section on line 259-260: “Briefly, purified recombinant SARS-CoV-2 antigens based on Wuhan Hu-1 isolate were coated on 96-well plates ...”

5. Comment 6 of R2 was not addressed sufficiently. The findings from ref 27 should be put in perspective with the results from the current study.

We have now reformulated this part of the Discussion, lines 224-232: “Recently, results of a clinical trial with 6 months follow-up on BNT162b2 vaccinees showed 86-100% vaccine efficacy against COVID-19 disease from 7 days after the second dose to the end of the surveillance period (27). They also showed vaccine efficacy of 100% in South Africa at the time when Beta variant was prevalent. Our results on BNT162b2 vaccine-induced neutralizing antibodies are well in line with the vaccine efficacy results, showing high neutralizing capacity of serum antibodies during interval of six weeks to 6 months post vaccination. The cell-mediated immunity contributes also to vaccine efficacy, and further analyses on T-cell immune responses in vaccinees are warranted”.

The reference 27 has a lot of appendix data, it being a 6-month follow-up report on phase 2–3 trial of BNT162b2 safety and efficacy, but we were unable to find any results on the analysis of neutralizing antibodies per se. We cannot make any direct relationships of neutralizing antibody levels and vaccine efficacy, since clinical vaccine efficacy is dependent on so many variables (time after vaccination, humoral and cell-mediated immunity, patients characteristics, circulating variants etc.). We think that how we have now stated the point raised by the Reviewer #2, is a good compromise what can be stated regarding neutralizing antibodies efficacy in relation to vaccine efficacy.

Reviewer comments:

Reviewer #1 (Comments for the Author):

My suggestions were sufficiently addressed to warrant publication. Figure 4 could be a supplemental figure, as it focuses on assay validation, but it's also fine to leave it in as a main figure.

We thank the Reviewer #1 for this positive comment. We would like to keep Fig 4 as a main figure because we feel it is very important to show what is the effect of variable parameters on MNTs. Many laboratories are using MNTs but with different parameters and this information on Fig 4 helps to compare MNT results from different countries/laboratories to get overall perspective on the neutralization capacities of sera from multiple sources.

February 6, 2022

Dr. Pinja Jalkanen
University of Turku
Institute of Biomedicine
Kiinamylynkatu 10
Turku 20520
Finland

Re: Spectrum02252-21R2 (Vaccine-induced antibody responses against SARS-CoV-2 variants-of-concern six months after the BNT162b2 COVID-19 mRNA vaccination)

Dear Dr. Pinja Jalkanen:

Your manuscript has been accepted, and I am forwarding it to the ASM Journals Department for publication. You will be notified when your proofs are ready to be viewed.

Sincerely,

Mathilde Richard
Editor, Microbiology Spectrum
